# A highly effective therapeutic ointment for treating corals with black band disease

Katherine R. Eaton[1,2,3]*, Abigail S. Clark[4,5], Katie Curtis[6], Mike Favero[6], Nathaniel Hanna Holloway[7,8], Kristen Ewen[7], Erinn M. Muller[1,4]

**1** Mote Marine Laboratory, Sarasota, FL, United States of America, **2** Cooperative Institute for Marine and Atmospheric Studies, University of Miami, Miami, FL, United States of America, **3** National Oceanographic and Atmospheric Administration (NOAA) Atlantic Oceanographic and Meteorological Laboratory, Ocean Chemistry and Ecosystems Division, Miami, FL, United States of America, **4** Elizabeth Moore International Center for Coral Reef Research and Restoration, Mote Marine Laboratory, Summerland Key, FL, United States of America, **5** The College of the Florida Keys, Key West, FL, United States of America, **6** Ocean Alchemists LLC, Tampa, FL, United States of America, **7** National Park Service, St. Croix, United States Virgin Islands, **8** Scripps Institution of Oceanography, La Jolla, CA, United States of America

* kre31@earth.miami.edu

**Data Availability Statement:** All relevant data are within the paper and its Supporting Information files.

## Abstract

Infectious disease outbreaks are a primary contributor to coral reef decline worldwide. A particularly lethal disease, black band disease (BBD), was one of the first coral diseases reported and has since been documented on reefs worldwide. BBD is described as a microbial consortium of photosynthetic cyanobacteria, sulfate-reducing and sulfide-oxidizing bacteria, and heterotrophic bacteria and archaea. The disease is visually identified by a characteristic dark band that moves across apparently healthy coral tissue leaving behind bare skeleton. Despite its virulence, attempts to effectively treat corals with BBD in the field have been limited. Here, we developed and tested several different therapeutic agents on *Pseudodiploria* spp. corals with signs of active BBD at Buck Island Reef National Monument in St. Croix, USVI. A variety of therapies were tested, including hydrogen peroxide-based treatments, ointment containing antibiotics, and antiviral/antimicrobial-based ointments (referred to as CoralCure). The CoralCure ointments, created by Ocean Alchemists LLC, focused on the dosing regimen and delivery mechanisms of the different active ingredients. Active ingredients included carbamide peroxide, Lugol's iodine solution, along with several proprietary essential oil and natural product blends. Additionally, the active ingredients had different release times based on treatment: CoralCure A-C had a release time of 24 hours, CoralCure D-F had a release time of 72 hours. The ointments were applied directly to the BBD lesion. Also, jute rope was saturated with a subset of these CoralCure ointment formulations to assist with adhesion. These ropes were then applied to the leading edge of the BBD lesion for one week to ensure sufficient exposure. Corals were revisited approximately three to five months after treatment application to assess disease progression rates and the presence/absence of lesions—the metrics used to quantify the efficacy of each treatment. Although most of the treatments were unsuccessful, two CoralCure rope formulations—CoralCure D rope and CoralCure E rope, eliminated the appearance of BBD in 100% of the corals treated. As such, these treatments significantly reduced the likelihood of BBD occurrence compared to the untreated controls. Additionally, lesions treated with these

**Funding:** Funding for this research was awarded to Dr. Erinn Muller from the National Park Service (https://www.nps.gov/index.htm) award # 240127. The funders had no role in study design, data collection and analysis, decision to publish, or preparation of the manuscript.

**Competing interests:** The authors have declared that no competing interests exist.

formulations lost significantly less tissue compared with controls. These results provide the mechanisms for an easily employable method to effectively treat a worldwide coral disease.

## Introduction

Coral disease outbreaks are one of the primary threats to coral reefs worldwide. One of the most prevalent and virulent coral diseases, black band disease (BBD), has been reported in coral reef regions all over the world [1–7]. BBD is characterized as a multispecies consortium of photosynthetic cyanobacteria, sulfate-reducing bacteria, sulfide-oxidizing bacteria, hetero-trophic bacteria and archaea [8–10]. *Roseofilum reptotaenium* is a common cyanobacterium of BBD within this microbial assemblage [11]. Proteobacteria and Bacteroidetes are also present in this assemblage, fueling the continued growth of *Roseofilum* by degrading and recycling organic matter within the mat [12]. This bacterial consortium presents as a dark black or red band, or mat, that moves across apparently healthy coral tissue, leaving behind exposed white skeleton [13].

While BBD is found globally [1–7], it has been routinely documented in the Caribbean region [2, 14, 15]. In fact, the Caribbean region is described as a coral disease "hotspot" [16] because over 70% of all reported coral diseases have occurred within this area since the early 2000s [1] and two of the worst coral disease outbreaks on record have occurred within this region [17, 18]. BBD affects over 42 species of Caribbean corals, including reef-building scler-actinian species as well as gorgonians [1, 2]. High mortality rates of entire coral colonies can occur during a BBD outbreak, and the disease is likely an important contributor to coral reef habitat degradation in the Caribbean [14, 19].

Environmental variables, such as water temperature and light, are often positively associated with BBD prevalence and virulence. In the Caribbean region, BBD typically occurs at shallower water depths (6 m or less) during times of high water temperature (29–30°C) [20, 21]. BBD lesions progress at a rate of approximately three mm/day, however rates of up to one cm/day have been documented [15]. Additionally, aquaria-based experiments have demonstrated that BBD progresses faster at elevated light (440 $\mu$mol m$^{-2}$ s$^{-1}$) [23], and temperatures (30–30.5°C) [22, 23]. Thus, developing effective treatments to halt the progression of BBD is critical for preserving coral communities and biodiversity as prevalence and virulence may become increasingly worse with climate change.

There have been several attempts to treat BBD *in situ*. The first proactive attempt to treat BBD occurred in the late 1980s - the BBD mats were removed from coral surfaces using an underwater aspirator, followed by sealing the disease interface with modeling clay [24]. Although this treatment appeared reasonably effective, with reinfection rates of only 30% (monitoring time period post treatment unknown), the approach was very labor-intensive [24]. Another study documented that creating a 'firebreak' within coral tissue and then treating the lesion with chlorinated epoxy was an effective treatment against BBD in *Montipora* spp., also with reinfection rates of 30% two months post-application [25]. However, this treatment has not been tested on BBD affecting Caribbean coral species. Additionally, the development of topical pastes mixed with antibiotics has been used to treat another lethal coral disease, stony coral tissue loss disease (SCTLD). Since this treatment appeared successful in halting the progression of SCTLD lesions, with success rates ranging from 67–95% depending on species [26, 27], attempting to utilize this treatment on other coral diseases, such as BBD, could be a viable option. Although, the use of antibiotics poses several concerns, such as

antibiotic resistance. Thus, natural product blends of antiviral and antimicrobial compounds should also be explored.

Concentrated (35%) hydrogen peroxide is a chemical that effectively kills cyanobacteria [28–30] and has been used by both aquaculture specialists and aquarists for many years [31, 32]. Additionally, corals naturally produce hydrogen peroxide in the wild as a bactericide to prevent pathogenic infection [33]. Since the BBD consortium is primarily dominated by cyanobacteria, concentrated hydrogen peroxide could also effectively treat BBD. Similarly, carbamide peroxide (hydrogen peroxide with urea) may also be an effective treatment for BBD. While powerful on contact, carbamide peroxide quickly degrades to water and oxygen making it less likely to accumulate in the surrounding environment and have impacts on any adjacent, non-infected coral communities. Another potential treatment for BBD is Lugol's iodine solution. Lugol's iodine solution is composed of elemental iodine and potassium iodide in water; often used as an antiseptic and disinfectant [34]. It is another tool used by aquaculture specialists and aquarists for antimicrobial treatment of pests and diseases [35, 36]. Also, iodine is an abundant micronutrient readily available in seawater [37], so antibacterial concentrations of iodine quickly dilute to the natural background concentration of iodine within oceanic waters [38]. However, the efficacy of using hydrogen peroxide, carbamide peroxide, or Lugol's iodine solution has not been tested in the field. Additionally, applying a liquid within a dynamic fluid environment such as shallow coral reefs poses logistical challenges because the liquid would need to be contained within an applicator and held against the treatment area to prevent immediate dilution (Erinn Muller, personal observations).

Here, we tested several different therapeutic agents on corals of *Pseudodiploria* spp. affected by BBD at Buck Island Reef National Monument in St. Croix, USVI, a location noted to have high prevalence of black band disease (Erinn Muller, personal observations). Therapeutic agents included hydrogen peroxide-based treatments, ointment containing antibiotics, and antiviral/antimicrobial-based ointments (referred to as CoralCure). We hypothesized that, because these treatments had shown some form of antimicrobial/antibacterial effects in other systems or in previous studies, each of the applications would effectively treat corals with BBD. The goals of the present study were to develop: i) a therapy to stop the progression of BBD, and ii) techniques for the effective and efficient application of this therapy *in situ*, which can ultimately be implemented across reefs within Buck Island Reef National Monument and potentially worldwide.

## Materials & methods

### Study site

All field work for this study was completed under permits VICR-2021-SCI-0003 and VIIS-2021-SCI-0015 issued by the National Parks Service. From July 2020—June 2021, 13 potential therapeutic agents and four control treatments were applied to corals of *Pseudodiploria* spp. showing signs of active BBD. The study took place at the northeast end of the shallow (approximately 2–5 m) lagoon of Buck Island Reef National Monument (**Fig 1**). Therapeutic agents were applied to the corals independently at one of four different time points: July 2020, October 2020, January 2021, or June 2021. The treatments applied at each time point are summarized in **Table 1**. The efficacy of these treatments was assessed at 3–5 months post treatment. Once the jute rope treatments appeared effective, a jute rope with no ointment was applied and included among the four control treatments. These applications were revisited one week after application (**Table 2**). A one-week post application survey was sufficient to determine whether BBD remained on the coral colony or not for this treatment control.

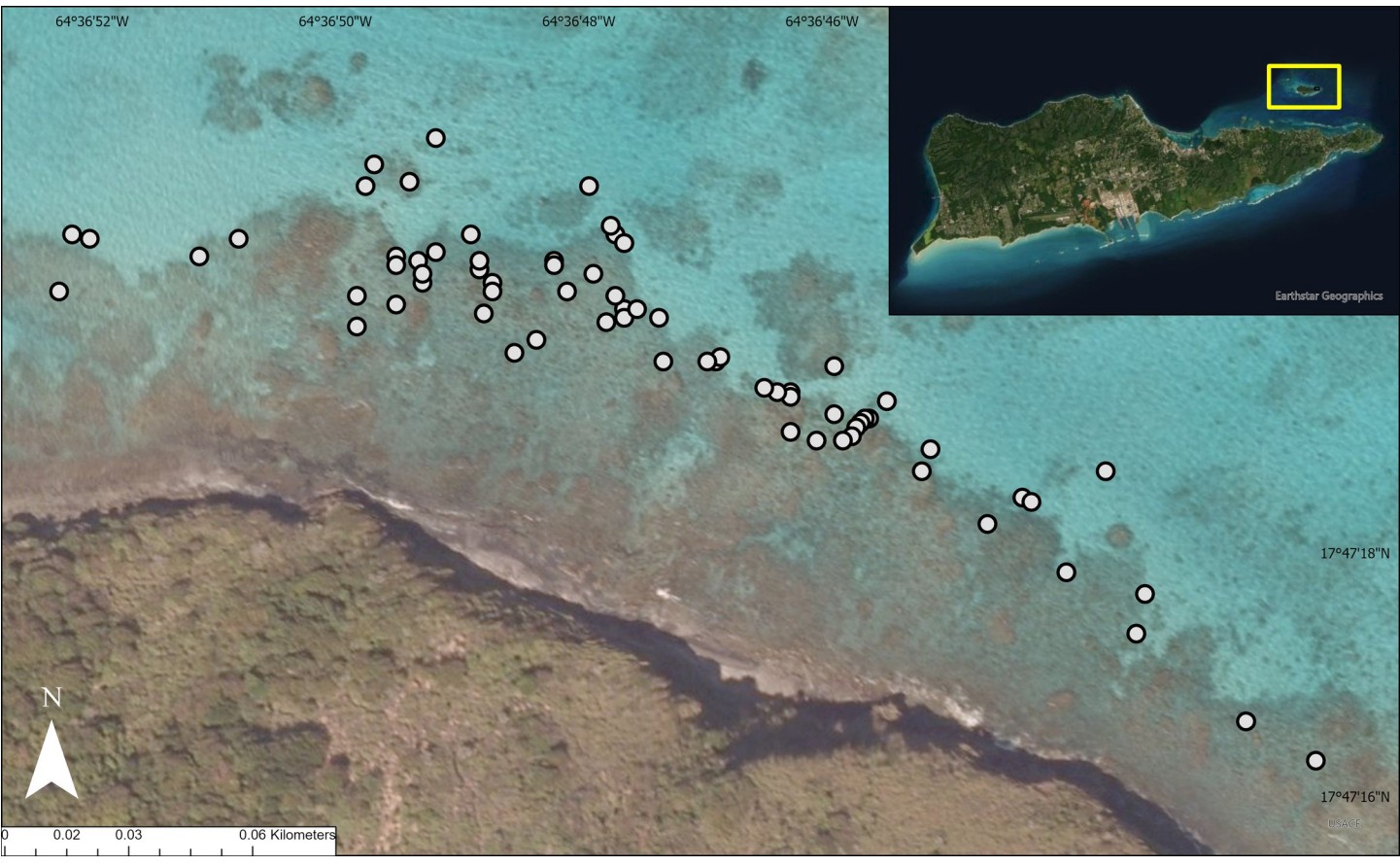

**Fig 1. Map of the study site (yellow box of the inset map), located at the Buck Island Reef National Monument in St. Croix, USVI.** The white circles represent colonies of *Pseudodiploria* spp. with BBD lesions (n = 72) that were tagged and used in this study.

To fate track colonies, each coral with a BBD lesion was assigned a unique identifier using a cattle tag attached to the dead part of the coral head with a masonry nail. After securing the cattle tag, the coral was photographed (Olympus Tough TG-6, Bethlehem, PA, USA) and measurements were taken *in situ* to later quantify disease progression. To measure progression rates of the BBD between time points, the distances between live tissue were measured at three equidistant points from the installed cattle tag nail (see **Fig 2** for example). Sketches of the coral with the respective locations of each measurement were conducted at the initial treatment period and were referenced when taking follow-up measurements to ensure that the same three angles from the nail were measured at each time point. The coral was then treated and, after approximately three months, revisited and re-measured.

## Therapeutic agent and application mechanisms

**Hydrogen peroxide-based treatments.** Three different hydrogen peroxide-based treatments were tested (n = 5 for each treatment, see **Table 1**). A prototype of an *in-situ* liquid applicator, consisting of flexible aquaria tubing (2 cm in diameter) cut in half lengthwise to create an open chamber and coated and secured with silicone glue, was created to deliver a direct application of 30% hydrogen peroxide to BBD lesions. A one-way valve was used to allow the injection of hydrogen peroxide from a 60 ml blunt tipped plastic syringe into the chamber and, as the hydrogen peroxide entered, a second valve allowed seawater to escape.

**Table 1. Summary of treatments applied to *Pseudodiploria* spp. colonies with BBD between July 2020 and June 2021.**

| Treatment | Sample Size (n) | Date of Treatment | Date of Revisit |
|---|---|---|---|
| $H_2O_2$ prototype applicator | 5 | July 2020 | October 2020 |
| $H_2O_2$ tooth whitening gel | 5 | July 2020 | October 2020 |
| $H_2O_2$ tooth whitening gel and Base2B | 5 | July 2020 | October 2020 |
| Base2B and amoxicillin | 5 | July 2020 | October 2020 |
| Epoxy only (control) | 5 | July 2020 | October 2020 |
| Base2B only (control) | 5 | July 2020 | October 2020 |
| Untreated (control) | 5 | July 2020 | October 2020 |
| CoralCure A ointment | 6 | October 2020 | January 2021 |
| CoralCure A rope | 5 | October 2020 | January 2021 |
| CoralCure B ointment | 6 | October 2020 | January 2021 |
| CoralCure B rope | 5 | October 2020 | January 2021 |
| CoralCure C ointment | 5 | October 2020 | January 2021 |
| CoralCure C rope | 3 | October 2020 | January 2021 |
| Base2B and amoxicillin | 5 | October 2020 | January 2021 |
| Untreated (control) | 5 | October 2020 | January 2021 |
| CoralCure D rope | 5 | January 2021 | June 2021 |
| CoralCure E rope | 5 | January 2021 | June 2021 |
| CoralCure F rope | 5 | January 2021 | June 2021 |
| Untreated (control) | 5 | January 2021 | June 2021 |

The prototype was filled with approximately 30 ml of hydrogen peroxide and held directly on the lesion for approximately one minute (**S1A Fig**). Additionally, a tooth whitening gel, Opalescence™ Boost™, containing 40% hydrogen peroxide (Opalescence™ Teeth Whitening Systems, Ultradent, South Jordan, UT, USA) was applied directly to the lesion and covered with marine epoxy (All-Fix Epoxy, Lafayette Hill, PA, USA) to help secure the gel to the coral (**S1B Fig**). In another treatment, the whitening gel was mixed with a delivery vehicle, Base2B (Ocean Alchemists LLC, Tampa FL, USA), which was applied directly to the lesion and anchored approximately every 5 cm with marine epoxy to hold the ointment in place (**S1C Fig**).

**Antibiotic-based treatment.** One antibiotic-based treatment, amoxicillin (PhytoTech Labs, Lenexa, KS, USA) mixed with Base2B (Ocean Alchemists LLC), was tested. Amoxicillin concentrations and application methods described in [26] were used for this study (see **S2 Fig** for application). The antibiotic-based treatment was applied in July 2020 (n = 5) and was repeated in October 2020 (n = 5), as this treatment appeared the most promising in July compared with others tested.

**CoralCure ointment and ropes.** CoralCure is a customizable, biodegradable ointment matrix (developed by Ocean Alchemists LLC) capable of carrying a wide range of active ingredients that may be effective in the treatment of coral disease. The ointment matrix can be formulated around various concentrations and release rates of the active compounds to

**Table 2. Summary of jute rope therapies and controls applied to *Pseudodiploria* spp. colonies with BBD in June 2021 and revisited one week post-treatment.**

| Treatment | Sample Size (n) |
|---|---|
| CoralCure D rope | 6 |
| Rope only (control) | 5 |
| Untreated (control) | 6 |

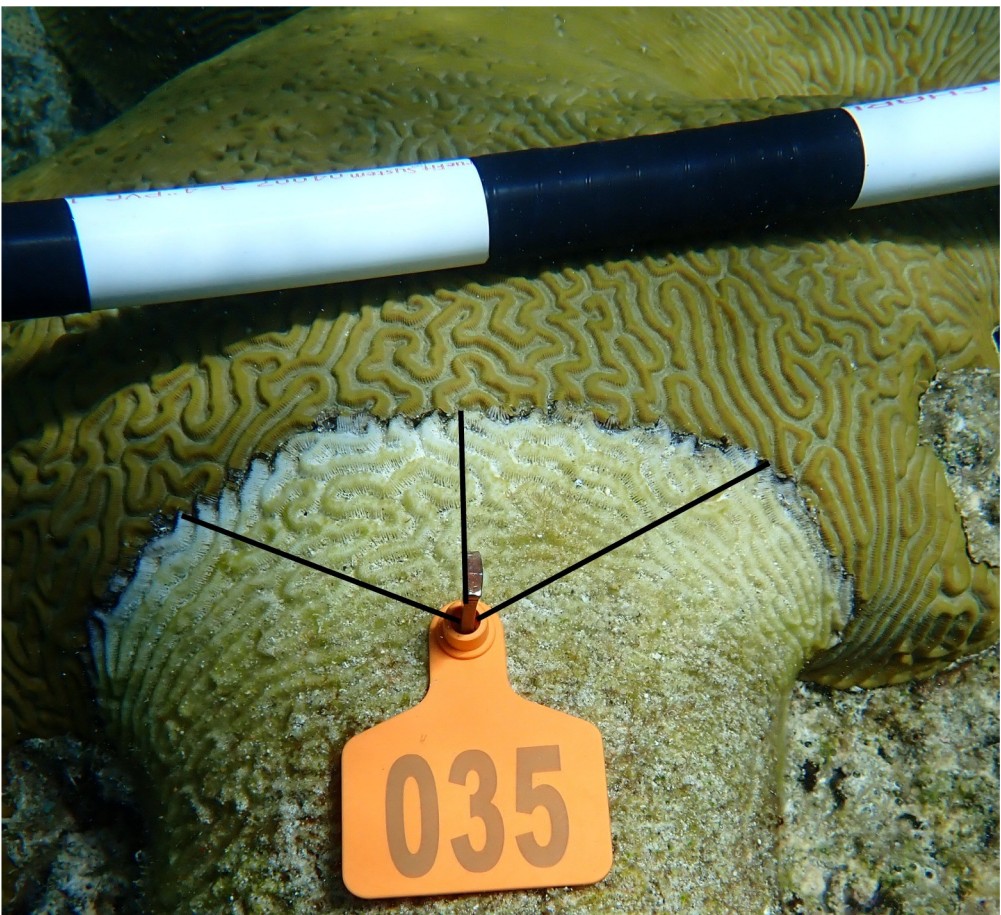

**Fig 2. Schematic representation of disease progression measurements collected from corals with active BBD.** The black lines represent three measurements taken (in cm) from the nail to the lesion of each colony. The cattle tag is used to identify and fate track the coral colony.

effectively dose against the targeted disease. Six CoralCure ointments were created for this study, CoralCure A—F (Table 3). Each of the ointments were designed utilizing a mucosal adhesive that allows the ointment to bind to the mucous membrane of living coral tissue. Ointments were then formulated around each active ingredient to stabilize and release the active compounds over a targeted dosing period. These treatment ointments were absent of antibiotics or living organisms which ensured their inability to contribute to antibiotic resistance or to mutate/proliferate.

CoralCure A was designed to deliver 15% Lugol's iodine solution. The iodine antiseptic solution from CoralCure A was then reduced in concentration and paired with essential oil and natural product extracts (#01) containing antibiofilm properties in CoralCure B. The added extracts were utilized in low concentrations, with the iodine antiseptic still acting as the primary active agent. These extracts are known to possess antibiofilm properties, which is a potentially important attribute as corals defend themselves against pathogens by regularly shedding a thick mucosal layer. This active ingredient combination was designed to help increase the effectiveness by assisting in penetration through the corals' defensive biofilm layer. CoralCure C utilized another broad-spectrum antiseptic in the form of carbamide peroxide, a stabilized precursor to hydrogen peroxide.

**Table 3. Description of CoralCure formulations tested against active BBD lesions on *Pseudodiploria* spp. colonies.**

| CoralCure | Primary Active Ingredient | Dosing Regime |
|---|---|---|
| A | 15% w/w Lugol's iodine solution | Paired with time release modifiers, 24-hour release time |
| B | 10% w/w Lugol's iodine solution, 5% w/w proprietary essential oil and natural product blend #01 | Paired with release modifiers, 24-hour release time. |
| C | 20% w/w carbamide peroxide | Approximately 24-hour release time. Note, the liberation of oxygen upon reaction significantly affected the ointment's ability to remain anchored in place. |
| D | 15% w/w proprietary essential oil and natural product blend #02 containing both antibacterial and antiviral properties | Paired with release modifiers, 72-hour release time |
| E | 15% w/w proprietary essential oil and natural product blend #03 containing both antibacterial and antiviral properties | Paired with release modifiers, 72-hour release time |
| F | 15% w/w proprietary essential oil and natural product blend #04 containing both antibacterial and antiviral properties | Paired with release modifiers, 72-hour release time |

*w/w represents weight of active ingredient / total weight of ointment.

CoralCure D, E & F each employed various combinations of proprietary essential oil and natural product blend blends (#02, #03, and #04) believed to work in a synergistic manner. Gram-negative cyanobacteria was the primary target [39], however natural products known to act against gram-negative bacteria, gram-positive bacteria, and viruses were combined in each formulation in an effort to eliminate the entire consortium of microbes that comprises BBD, as well as any putative pathogens that may target vulnerable coral tissue. These blends utilized a 72-hour release rate, which is similar to the amoxicillin/Base2B treatment.

CoralCure A—C ointments were the first of the CoralCure formulations tested. These formulations were applied directly to the BBD lesion and into approximately 2–3 cm of the apparently healthy tissue adjacent to the lesion (S3 Fig). Additionally, jute ropes were used as a delivery vehicle for these formulations, as well as formulations D—F. Jute ropes were heavily coated with the ointment and applied to the coral using a heavy-duty staple gun (Arrow Fastener Co., LLC, Saddle Brook, NJ, USA). Ropes were 1.5875 cm wide, and the lengths of rope depended on the size of the lesion. For CoralCure A—C, one rope was applied directly to the BBD lesion (Fig 3A and 3B). Due to the lack of adhesion, CoralCure D—F ointments alone were not tested, only the jute rope saturated with ointments. For CoralCure D—F, two ropes were applied side-by-side directly to the BBD lesion and into the apparently healthy tissue adjacent to the lesion (3.175 cm wide total; Fig 3C and 3D). Two ropes were applied for Coral-Cure D—F to increase the area of tissue treated and, in so doing, treat the BBD lesion before it progressed and reached the other side of the rope (closest to the apparently healthy tissue). Ropes and staples were removed from the coral after one week of exposure.

**Control conditions.** Four control treatments were applied (see **Tables 1 and 2**). Since marine epoxy, Base2B, and jute ropes were used as part of the delivery process for the active treatments, these were applied to the corals by themselves as well. The control epoxy and Base2B treatments were applied the same way as described for the active treatments (see S4A and S4B Fig). However, the control jute ropes were secured to the corals using nails instead of staples because without the ointment coating the jute rope, loose jute fibers got trapped in the staple gun and prevented it from functioning properly (S4C Fig). The jute rope and nails were removed after one

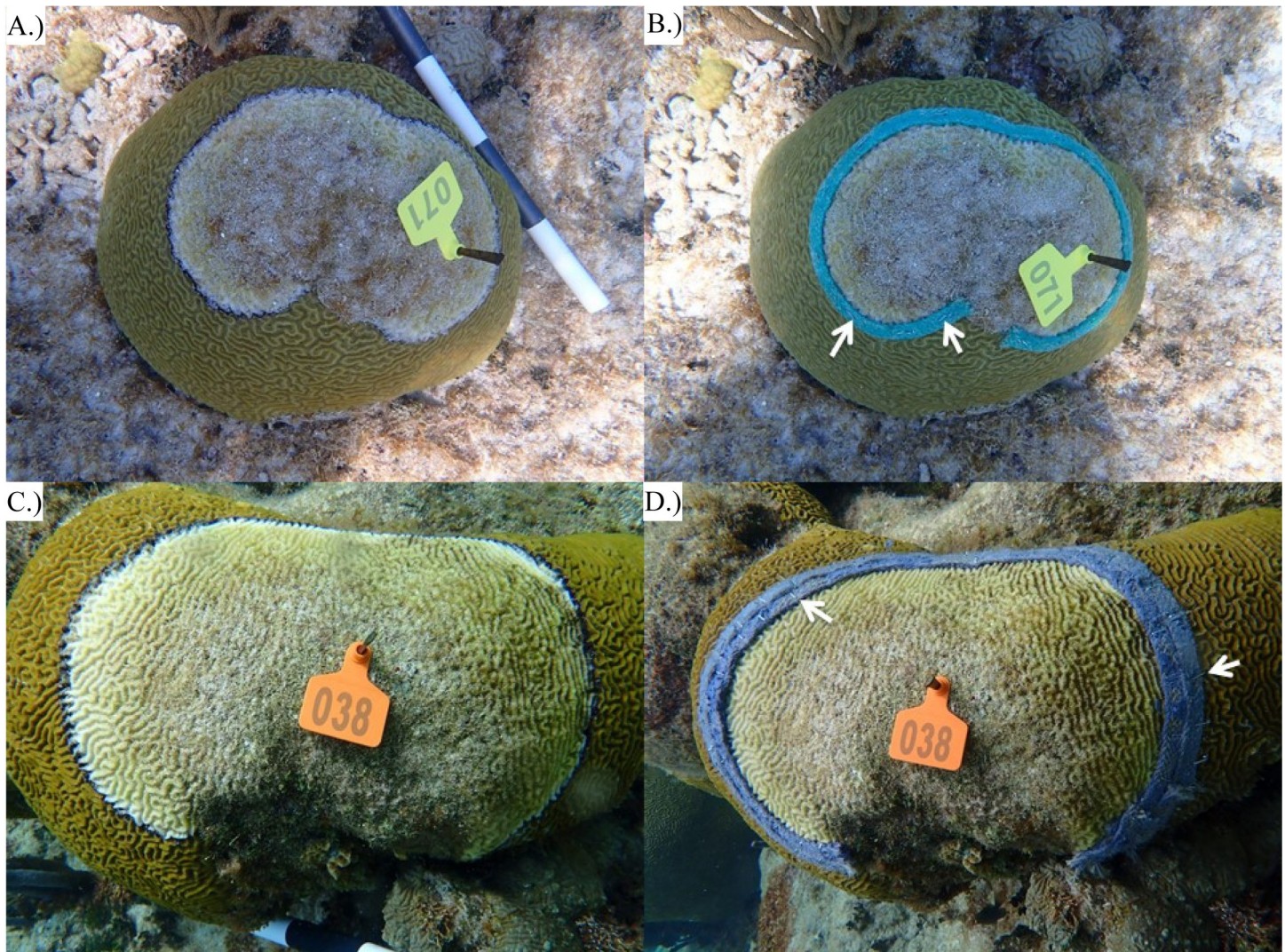

**Fig 3. Representative photos of *Pseudodiploria strigosa* colonies treated with ropes saturated in CoralCure.** (**A**) Colony #71 black band disease (BBD) lesion before treatment application, (**B**) A single CoralCure A rope applied to colony #71 BBD lesion, (**C**) Colony #38 BBD lesion before treatment application, (**D**) Two CoralCure D ropes applied side-by-side to colony #38 BBD lesion. The white arrows indicate a subset of staples used to hold the ropes in place.

week of exposure. Control jute ropes were applied in June 2021 and revisited only after one week (vs. several months; see **Table 2**) because were applied during the final field trip of the study. Therefore, only BBD presence/absence data were collected after rope removal one week post-application and disease progressions rates were unavailable for this control treatment. Additionally, untreated controls were set up every time a new treatment was implemented (**S4D Fig**). Untreated controls consisted of colonies with BBD that were not treated with any therapeutic agent nor application control. There were no CoralCure ointment controls within the study.

## Coral monitoring

Corals were revisited three to five months post-treatment (except the rope controls that were surveyed one week post-treatment at the end of the study) to calculate the efficacy rates of each treatment, which were determined by disease progression rates (i.e., loss of coral tissue) and the

presence/absence of active BBD lesions. The distance between the live tissue and the masonry nail was again measured at the same three equidistant points from the nail within each colony when the corals were revisited (Fig 2). For the rope treatments, the width of the rope (1.5875 cm for single rope on CoralCure A—C, 3.175 cm for double wide rope for CoralCure D—F) was subtracted from the three measurements to account for the area that the rope covered and that had tissue loss as a direct result of the treatment application. The average difference in the distance of live tissue between surveys quantified the linear progression rates of the BBD. If the treatment was effective, then little to no tissue loss should have occurred after treatment. Additionally, during the three to five month post-treatment assessment, it was noted whether BBD was visually apparent and active. Corals were photo documented during the post-treatment surveys.

## Data analysis

Statistical analyses were performed in the program R [40], and figures were made using Prism (version 8.0.0 for Windows, GraphPad Software, San Diego, California USA). One-way analysis of variance (ANOVA) tests were used to assess the statistical significance of disease progression between treated and control lesions. Data not normally distributed were analyzed using a Kruskal-Wallis test. A Fisher's Exact test [41] was used to determine the likelihood of there being active BBD lesions present in treated and untreated control corals three months post-treatment. All variations are reported as standard error of the mean.

## Results

### July 2020, October 2020 treatments

**Disease progression.**   There were no statistical differences detected between the lesion progression rates of BBD lesions treated with any $H_2O_2$ therapies, antibiotics, or CoralCure A—C (ointment and jute rope applications) compared with respective controls (S1 Table). Corals treated with $H_2O_2$-based treatments had an average disease progression of 2.49 ± 0.30 cm/month; corals treated with antibiotics had an average disease progression of 1.91 ± 0.96 cm/month (Fig 4). Corals treated with CoralCures A—C (ointment and jute rope applications) had an average disease progression of 1.21 ± 0.43 cm/month, 1.08 ± 0.53 cm/month, and 0.82 ± 0.39 cm/month, respectively (Fig 5).

**Presence/absence of lesion upon revisit.**   There were no statistical differences in the presence/absence of BBD lesion results for treatments applied in both July 2020 and October 2020 (S2 Table). Therefore, a similar number of corals had active BBD lesions in the post-treatment monitoring surveys for both July 2020 and October 2020 treatment applications, regardless of whether they were treated or control corals.

### January 2021 treatments

**Disease progression.**   There was a significant difference in lesion progression of Coral-Cure D rope-treated lesions compared with untreated lesions (Fig 6, Kruskal-Wallis, H(1) = 3.872, p = 0.049). Untreated control lesions progressed significantly faster (0.64 ± 0.19 cm/month) than lesions treated with CoralCure D ropes (0.26 ± 0.04 cm/month) over the five-month period. Additionally, there was a significant difference in lesion progression of Coral-Cure E rope-treated lesions compared with untreated control lesions (Fig 6, Kruskal-Wallis, H(1) = 4.86, p = 0.027). Untreated control lesions progressed significantly faster (0.64 ± 0.19 cm/month) than lesions treated with CoralCure E ropes (0.22 ± 0.09 cm/month). There was no significant difference between the lesion progression of CoralCure F rope-treated lesions and untreated control lesions (Fig 6, Kruskal-Wallis, H(1) = 1.83, p = 0.176). One untreated

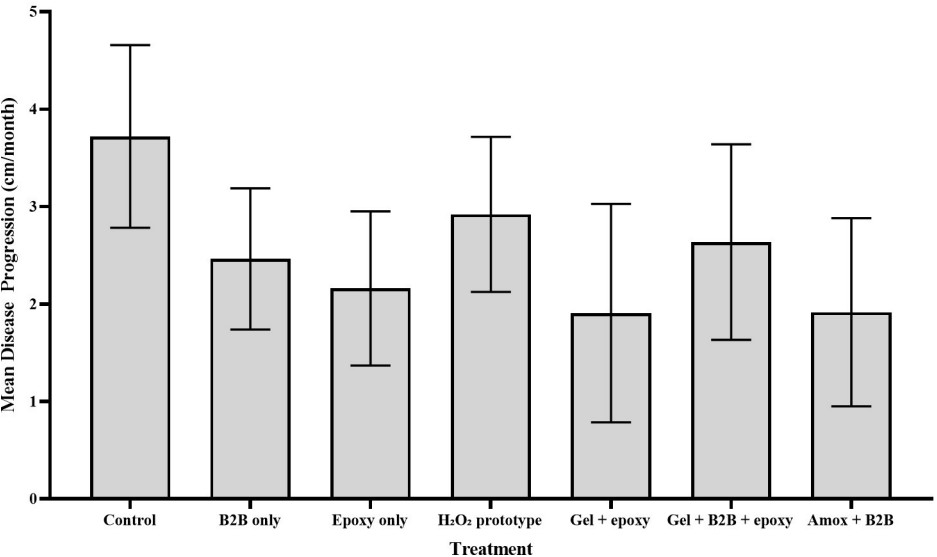

**Fig 4. Mean progression (cm/month) of BBD in *Pseudodiploria* spp. exposed to hydrogen peroxide ($H_2O_2$)-based treatments [i.e., prototype, tooth whitening gel (gel) with and without marine epoxy and Base2B] or the Base2B (B2B) and amoxicillin (amox) treatment in July 2020 and the associated controls.** Corals were revisited in October 2020. Error bars represent standard error of the mean.

control coral was unable to be located at the time of revisit and was therefore not included in the analysis.

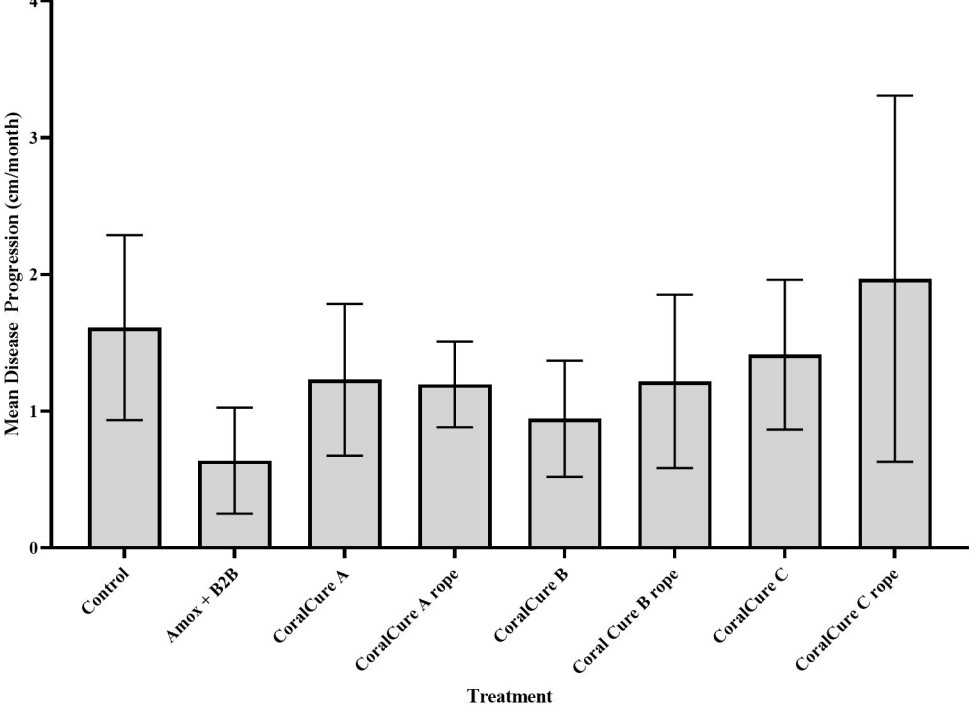

**Fig 5. Mean progression (cm/month) of black band disease (BBD) in *Pseudodiploria* spp. exposed to CoralCure A—C ointments, their respective rope treatments, or the Base2B (B2B) and amoxicillin (amox) treatment in October 2020.** Also shown is the mean progression of BBD in untreated (control) corals. Corals were revisited in January 2021. Error bars represent standard error of the mean.

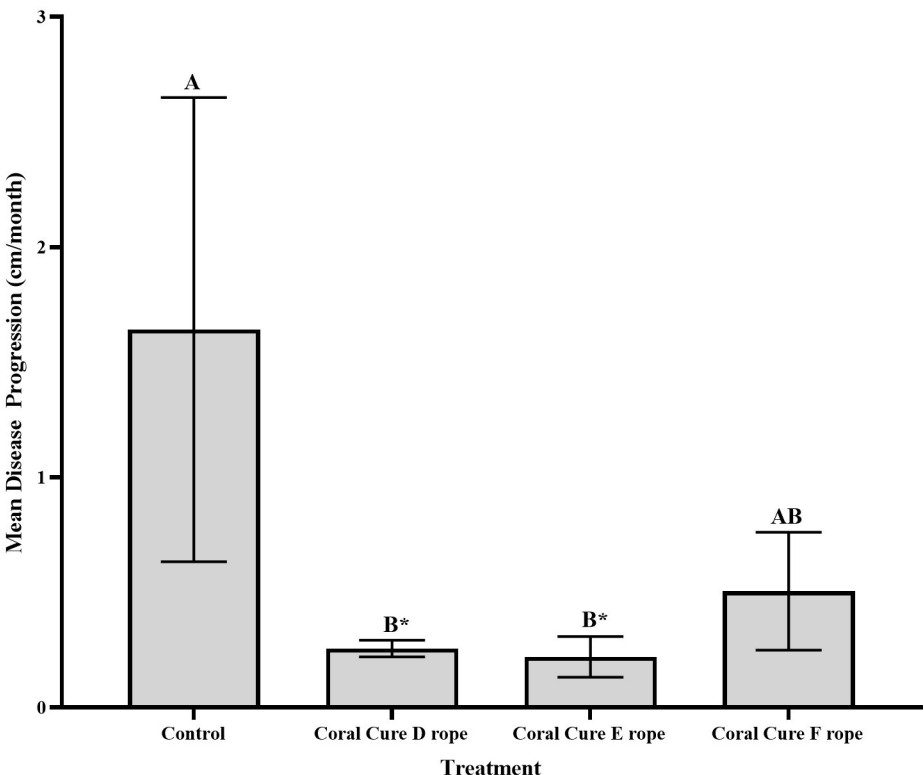

**Fig 6. Mean disease progression of *Pseudodiploria* spp. treated with CoralCures D (n = 5), E (n = 5), and F ropes (n = 5), and untreated controls (n = 4).** Error bars are standard error of the mean, and letters denote significant differences. Disease progression was calculated as the difference in the amount of tissue loss that occurred from January—June 2021 averaged by month.

**Presence/absence of lesion upon revisit.** Corals treated with CoralCure D rope and CoralCure E rope did not have active BBD lesions at the time of revisit. Three out of four untreated control corals had active BBD lesions. There was a significant difference in probability of lesion appearance for corals treated with CoralCure D rope and CoralCure E rope when compared to controls (Fisher's Exact, p = 0.047 for both treatments). One coral treated with CoralCure F rope had an active BBD lesion at the time of revisit, and there was not a significant difference in probability of lesion appearance compared to controls (Fisher's Exact, p = 0.206). Results are summarized in **Table 4**.

## June 2021 treatments

**Presence/absence of lesion upon revisit.** Zero out of six (0%) lesions treated with CoralCure D rope in June 2021 were active after rope removal (i.e., after one week). Six out of six

**Table 4. Presence/absence of active BBD lesions for each treatment upon revisit.**

| Date of treatment | Treatment | No. of treated corals with active BBD lesions at revisit (n = 5) | No. of control corals with active BBD lesions at revisit (n = 4) | Statistical values (Fisher's Exact) |
|---|---|---|---|---|
| January 2021 | CoralCure D rope | 0 | 3 | p = 0.047 |
| January 2021 | CoralCure E rope | 0 | 3 | p = 0.047 |
| January 2021 | CoralCure F rope | 1 | 3 | p = 0.206 |

(100%) jute rope control lesions remained active at the time of rope removal (see **Fig 7** for example). CoralCure D rope treatments significantly reduced the likelihood of the corals showing signs of BBD compared to the untreated controls (Fisher's Exact, p = 0.008) and jute rope controls (Fisher's Exact, p = 0.002). Five out of five (100%) untreated control lesions remained active after one week. Results are summarized in **Table 5**. **Fig 8** represents a time-series comparison of a CoralCure D rope-treated coral and an untreated control coral.

## Discussion

The present study is the first to use natural product extracts to treat a coral disease using an easy to employ method within the natural environment, and the first to show the potential for 100% treatment efficacy for a worldwide coral disease. Coral disease treatment has been successfully used for a few coral diseases around the world [24, 26, 27, 42–44], but none with such

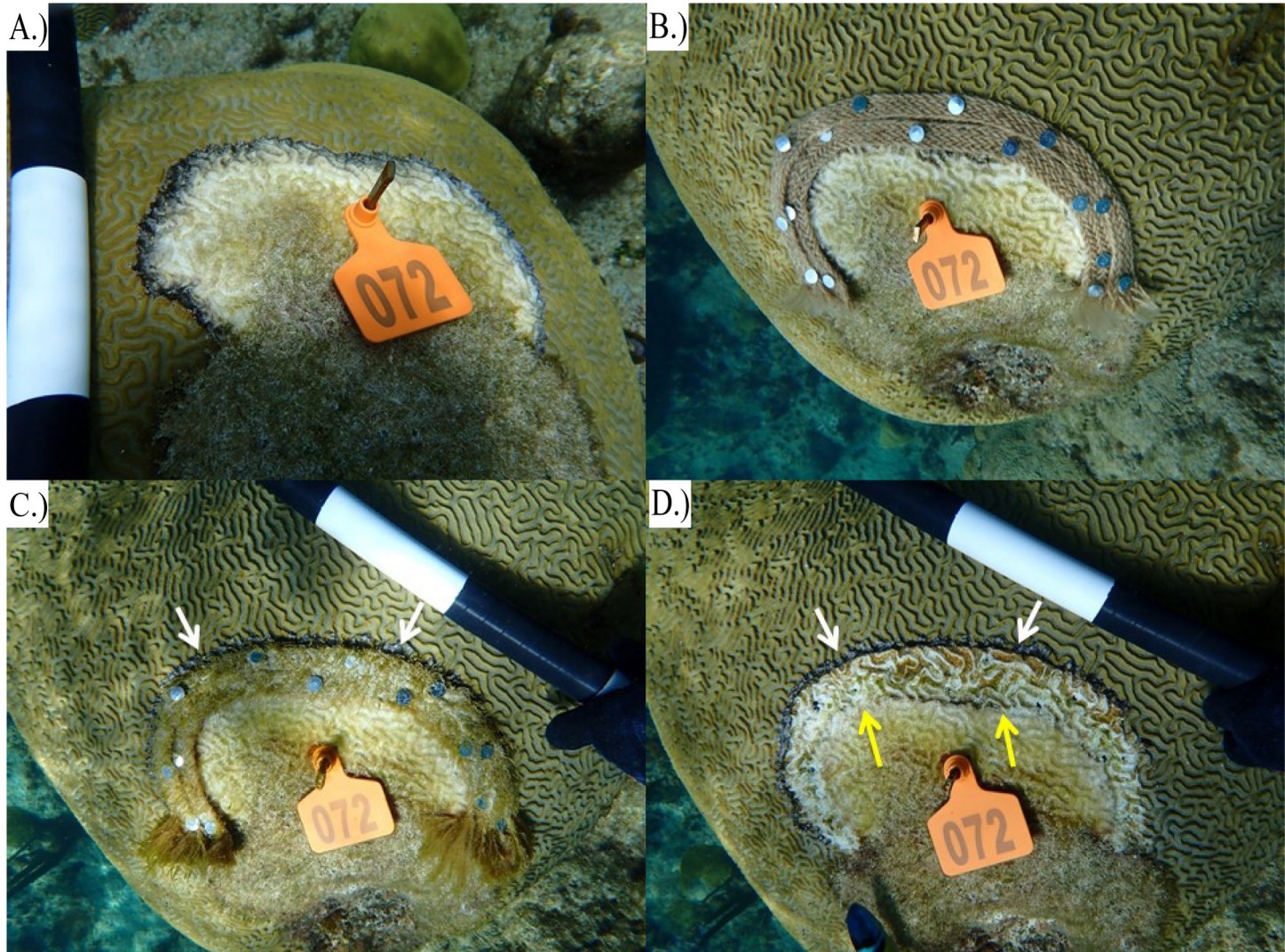

**Fig 7. Representative jute rope control.** *Pseudodiploria strigosa* with **(A)** an active BBD lesion prior to jute rope application, **(B)** two jute ropes applied to BBD lesion and apparently healthy tissue adjacent to lesion, **(C)** the same two ropes one week post-application (arrows denote BBD lesion moving under ropes), and **(D)** BBD lesion after rope removal (the white arrows indicate the location of the BBD lesion after rope removal, and the yellow arrows indicate the location of the BBD lesion at time of treatment).

**Table 5. Presence/absence of active BBD lesions for each treatment upon revisit.**

| Date of treatment | Treatment Comparison | No. of treated corals (n = 5) with observable BBD at revisit | No. of control corals (n = 5) with observable BBD at revisit | Statistical values (Fisher's Exact) |
|---|---|---|---|---|
| June 2021 | CoralCure D rope vs. control | 0 | 5 | p = 0.008 |
| June 2021 | CoralCure D rope vs. jute rope control | 0 | 6 | p = 0.002 |

high efficacy rates. Out of the 13 treatments tested in the present study, the CoralCure D and E rope treatments were the most effective. All corals treated with CoralCure D and E rope treatments remained apparently healthy—there were no active lesions at the time of revisit several months later. Lesions treated with CoralCure D rope in January 2021 progressed significantly slower than the untreated controls, only 0.26 ± 0.04 cm/month compared with 0.64 ± 0.19 cm/month respectively. Additionally, lesions treated with CoralCure E rope progressed significantly slower than the untreated controls (0.22 ± 0.09 cm/month compared with 0.64 ± 0.19 cm/month respectively). Notably, the active ingredients within these CoralCure formulations had a 72-hour release time, whereas CoralCure A—C had a 24-hour release time and did not appear successful. Although the active ingredients within CoralCure D and E were different, it is possible that longer dosing regimens (i.e., 72-hour) may be necessary for BBD treatment success. More studies are needed to further examine the relationship between release time and BBD progression.

There are two other studies that were successful in treating BBD, with reinfection rates of 30% [24, 25], but our method is extremely less labor intensive compared to [24]'s underwater aspirator method and more logistically feasible to employ at large spatial scales. It is possible that the method described in [25], chlorinated epoxy, would be successful on Caribbean brain corals, but this method has not yet been tested on any Caribbean species with BBD. Interestingly, the chlorinated epoxy treatment required an initial aspiration of the black band disease, an application of chlorinated epoxy along the lesion, as well as one applied as a firebreak 2–5 cm into apparently healthy tissue. The CoralCure D and E ropes in the present study were also applied as two ropes, but these were placed side by side to reduce the probability of the motile cyanobacteria of the black band consortia breaking through the treatment application area. Ultimately, this reduced the amount of healthy tissue lost to the treatment application itself. Treated corals in the present study showed only a 0.2 cm/month progression rate, which was three times less than the control corals—a result comparable to the [25] study. This limited loss of tissue, also, likely occurred soon after treatment as the corals appeared completely healthy (i.e., had no signs of recent tissue loss) several months later during the monitoring event.

Although more time was necessary for applying two ropes to the BBD and further time required for rope removal, when comparing with previous application methods such as aspiration/epoxy [24] and aspiration/firebreak/chlorinated epoxy [25], the double layer of saturated jute rope only took approximately five minutes per coral to apply and approximately two minutes to remove. Although no other previous studies document time necessary for treatment, aspiration itself (i.e., the removal of the black band) can be time consuming, creates effluent that must be contained and removed from the reef, and could create flocculates of the mat as it is disturbed during aspiration process, which could inadvertently cause transmission to neighboring corals. If treatment sites are visited frequently, the presently proposed application would be a highly efficient and comparatively quick method for effectively eliminating BBD within affected corals.

Since BBD is characterized as a multispecies consortium of photosynthetic cyanobacteria, sulfate-reducing bacteria, sulfide-oxidizing bacteria, heterotrophic bacteria and archaea

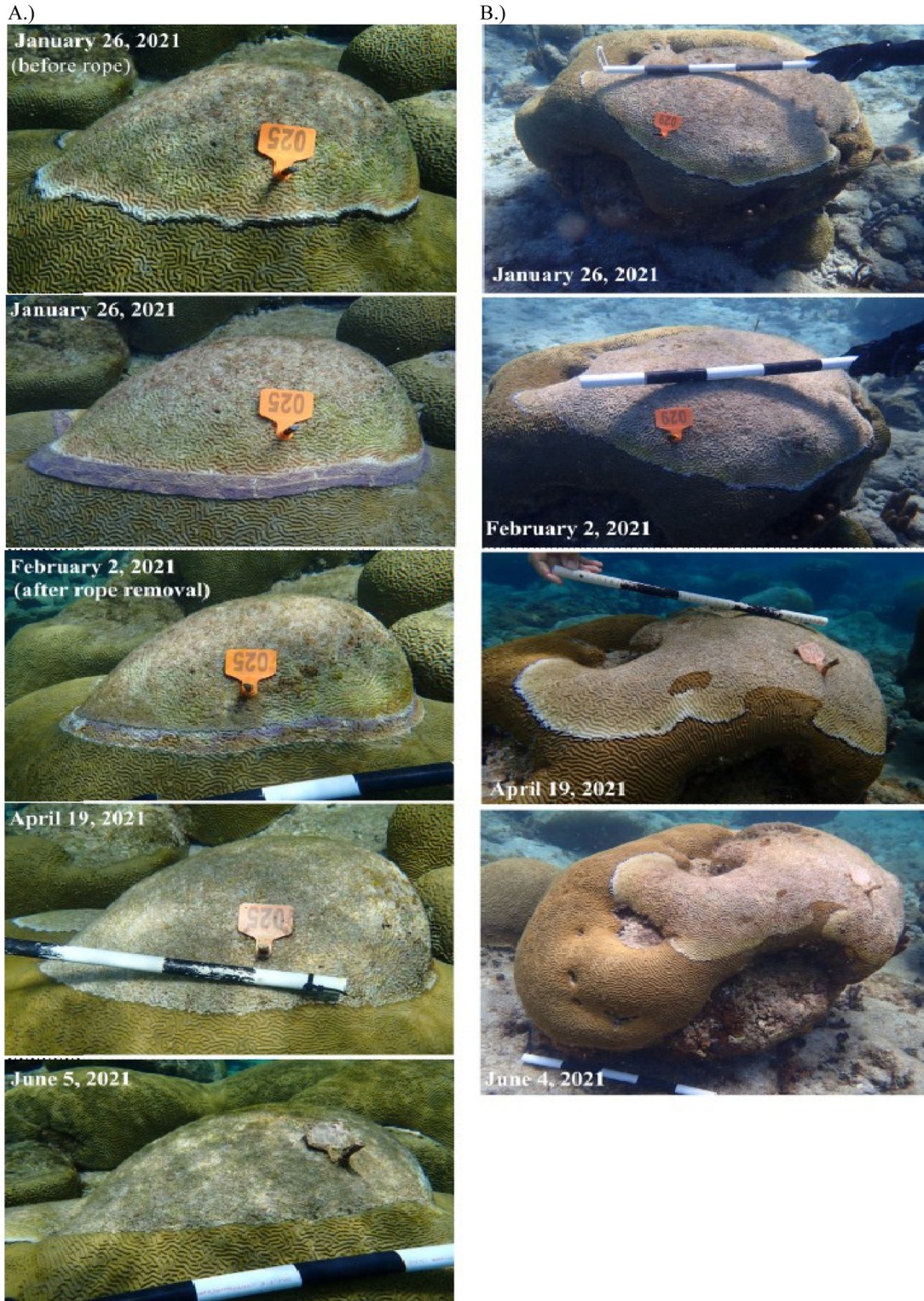

**Fig 8.** Time-series photographs of (A) a *Pseudodiploria strigosa* colony with active black band disease (BBD) treated with CoralCure D rope and of (B) an untreated *P. strigosa* with BBD. The black and white bars are each 10 cm in length.

[8–10], utilizing a broad-spectrum of active ingredients were considered to target several of the members of the consortium. Thus, the most effective treatments within the present study utilized a combined approach that included active ingredients that targeted gram-negative bacteria, gram-positive bacteria, as well as viruses to fight against this consortium of BBD bacteria and any opportunistic organisms that may attack the impaired tissue. It is important to note however that the active ingredients are only a piece of the puzzle towards designing an effective treatment method. Factors such as dosing regimen, ointment matrix composition, application method, species of coral tested, tissue targeted, and oceanic conditions all play pivotal roles in the treatment's failure or success. Additionally, the ointment without any active ingredients was not tested as a comparative control within the study. Next steps should include ointments with single active ingredients as well as those without any active ingredients. Each of the treatment methods utilized should also be further investigated for their potential impacts on the surrounding ecosystem. While many of the ingredients utilized in this study are widely available, it would be ineffective and potentially dangerous for them to be applied without the correct delivery mechanism or proper permitting.

There were several challenges associated with implementing the 13 different BBD treatments *in situ*, which could have been why most were unsuccessful. For example, although concentrated (35%) hydrogen peroxide ($H_2O_2$) is a chemical that effectively kills cyanobacteria [28–30], and preliminary data suggested that a single application of hydrogen peroxide efficiently stops BBD progression *ex situ* (Muller, personal observations), the hydrogen peroxide treatments tested here were unsuccessful. Upon application of hydrogen peroxide via the prototype applicator, bubbles immediately formed in the treated area (indicating the hydrogen peroxide was producing a reaction, see **S5 Fig**), but quickly dissipated. The rapid formation of oxygen bubbles was also observed with the hydrogen peroxide-based ointments, which resulted in the ointment often sloughing off the lesion within seconds to minutes after application. Therefore, the exposure time was likely too short to effectively kill the BBD microbial consortium.

Adhesion of the treatments to the corals was one of the biggest challenges. Since BBD is typically found in shallow waters [20], our chosen study site was in a shallow lagoon with constant surge, making the application of disease treatments challenging and potentially risky to the diver. The hydrogen peroxide treatments with a delivery mechanism (tooth whitening gel and tooth whitening gel/Base2B), amoxicillin/Base2B treatment, and CoralCure A—C ointments did not effectively adhere to the lesion or the apparently healthy tissue adjacent to the lesion (see **S6 Fig**). Since the amoxicillin/Base2B has been successful in treating SCTLD, [26, 27], it is possible that the lack of adhesion is why it did not work in the present study. The Base2B delivery vehicle has primarily been used in deeper areas (5 m—13 m) [26, 27], where surge is less prevalent. Additionally, the Base2B adheres well to recently exposed coral skeleton without any fouling organisms. Because BBD has an active mat of microorganisms in between the apparently healthy tissue and lesion, the adhesion may have been hindered by the absence of an exposed skeleton. Regardless, the challenges associated with adhesion prevented the active ingredients from interacting with the BBD for prolonged periods of time for many of these applications.

To overcome adhesion issues, jute rope was saturated in CoralCure ointments D—F. The use of jute rope as a delivery vehicle provided several advantages over the standard ointment delivery mechanisms. Physically attaching the ointment-covered rope to the coral ensured adhesion of the ointment to coral. However, attaching two ropes to the coral (one to the lesion and one to the apparently healthy tissue adjacent to the lesion) was necessary for success. When only one rope was applied to the lesion, such as for CoralCure ropes A—C, the BBD lesion progressed beyond the area covered by the rope (**S7 Fig**). When applying two ropes to

the affected area, the additional rope killed the apparently healthy tissue adjacent to the lesion. However, use of two ropes prevented the lesion from progressing beyond the treated region. Additionally, sunlight promotes growth of the BBD microbial consortium [22, 23], and the jute rope method blocked the sunlight from reaching the lesion, potentially slowing the growth of the microbial community.Although using the ointment saturated jute rope method was effective for treating BBD lesions, there are some disadvantages of using this method *in situ*. Using a staple gun to attach the rope to the corals underwater could cause injury to the diver if not applied carefully, especially in areas where surge is common (typically in shallow waters where BBD is most prevalent). Additionally, using a staple gun with jute rope may take slightly longer than the standard ointment application, however, when the ointment didn't stick and reapplication was routinely tried, the rope application was faster and more efficient. Unlike with ointments where, in theory, can be allowed to dissipate over time, the rope method needed two site visits. The first visit was to apply the rope and the second was a week later to remove the rope, which added considerably more resources necessary for this application method. An additional consideration is the rope method often required two people for the application—one to hold the rope on the appropriate area of the coral and one to staple the rope onto the coral. Rope applications also produced more waste than the standard ointment application, as ropes and staples were removed from the corals after one week and discarded. These staples, however, were easy to remove and did not cause additional harm to the coral beyond the ointment. Furthermore, the staple guns required regular maintenance and occasionally needed to be replaced. Lastly, this treatment does not necessarily stop the potential of the coral developing new lesions outside of the treated area, however, this only occurred in two out of five corals treated with CoralCure D rope and zero out of five corals treated with CoralCure E rope at the five-month revisit. Lesion development outside of the treated area was not quantified by either [24] or [25], although this has happened in the treatment of SCTLD [26, 27]. Additionally, some studies suggest that BBD lesions can stop or slow down in the winter months but become active again in the summer due to the seasonal increase in temperature and light [23]. Thus, for this treatment to be effective long-term, the corals must be re-visited and re-treated if new lesions develop.

## Conclusions

In the present study, a total of 13 different therapeutic agents were tested on *Pseudodiploria* spp. corals with active BBD lesions at the Buck Island Reef National Monument, St. Croix, USVI. Two of the 13 formulations halted lesion appearance and slowed progression in 100% of corals treated, with CoralCure D and E rope treated lesions progressing significantly slower than untreated controls and rope controls. Results indicate dosing release rate to be a critical area of further research as all treatments capable of dosing for ~72 hours achieved varying levels of success. Although these treatments only address the signs and not the underlying cause of BBD, they may be useful in containing BBD outbreaks and ultimately preserving coral communities and biodiversity. Additionally, there is applicability to manage this disease in a strategic way. BBD often only occurs seasonally [14, 45], and moves through the water in shallow areas, providing easy access and more time for divers to treat corals. Therefore, it is possible to develop management strategies to treat initial infections in the spring or early summer and potentially reduce subsequent intra- and inter-colony transmission. For example, to manage this disease at Buck Island, one round of treatments could be applied in early summer, with routine monitoring visits occurring monthly thereafter. Monitoring visits could be reduced throughout the winter. This method of treating BBD may be more applicable at large spatial scales compared to treating SCTLD, as treating SCTLD is persistent, affects corals at large

depth ranges, and retreatment is often necessary [26, 27]. Although there is always the possibility for new lesions to develop within a colony and reinfection of inactive lesions, the results presented here provide evidence for an easily employable method to effectively treat a worldwide coral disease. As such, these disease treatments should be considered by global resource managers as they continue to address BBD and other coral disease outbreaks.

## Supporting information

**S1 Fig. Hydrogen peroxide ($H_2O_2$)-based treatments tested. (A)** $H_2O_2$ prototype being held over the black band disease (BBD) lesion, **(B)** Tooth whitening gel (not visible) applied to a BBD lesion and held in place with marine epoxy (white), **(C)** Tooth whitening gel (red ointment) mixed with Base2B applied to a BBD lesion, and anchored in place with marine epoxy (white).
(TIF)

**S2 Fig. Amoxicillin mixed with Base2B (white) applied to a black band disease (BBD) lesion.**
(TIF)

**S3 Fig. Representative photo of CoralCure ointment applied to a BBD lesion of *Pseudodiploria strigosa* and apparently healthy tissue adjacent to the lesion.**
(TIF)

**S4 Fig. The four control treatments applied to *Pseudodiploria strigosa*. (A)** Marine epoxy covering the BBD lesion, **(B)** Base2B applied to a black band disease (BBD) lesion, **(C)** Two pieces of jute rope covering the BBD lesion and apparently healthy tissue adjacent to the lesion, **(D)** Untreated control lesion.
(TIF)

**S5 Fig. Example of hydrogen peroxide bubbles forming in the treated area, using the hydrogen peroxide prototype applicator.**
(TIF)

**S6 Fig. Example of CoralCure A ointment not effectively adhering to the BBD lesion.**
(TIF)

**S7 Fig. Example of a BBD lesion migrating under a CoralCure C rope (photo taken one week after rope application).** Yellow arrows represent where the BBD lesion started; white arrows indicate where the lesion migrated beyond the rope.
(TIF)

**S1 Table. Summary of results from one-way analysis of variance (ANOVA) and Kruskal-Wallis tests comparing the progression of BBD under different treatments.** Treatments were applied to *Pseudodiploria* spp. colonies in July 2020 and October 2020.
(DOCX)

**S2 Table. Presence/absence of active BBD lesions for each treatment upon revisit.**
(DOCX)

## Acknowledgments

We would like to acknowledge Anna Toline, Zandy Hillis-Starr, and Cheryl Woodley for assistance with experimental design and treatment development. We would like to thank Monty Clark for the $H_2O_2$ prototype development and Jessyca Garlock for assistance in the field.

## Author Contributions

**Conceptualization:** Erinn M. Muller.

**Data curation:** Katherine R. Eaton, Abigail S. Clark, Erinn M. Muller.

**Formal analysis:** Katherine R. Eaton, Abigail S. Clark, Erinn M. Muller.

**Funding acquisition:** Erinn M. Muller.

**Investigation:** Katherine R. Eaton, Abigail S. Clark, Katie Curtis, Mike Favero, Nathaniel Hanna Holloway, Kristen Ewen, Erinn M. Muller.

**Methodology:** Katherine R. Eaton, Abigail S. Clark, Katie Curtis, Mike Favero, Erinn M. Muller.

**Project administration:** Nathaniel Hanna Holloway, Kristen Ewen, Erinn M. Muller.

**Resources:** Katie Curtis, Mike Favero, Nathaniel Hanna Holloway, Kristen Ewen.

**Supervision:** Erinn M. Muller.

**Writing – original draft:** Katherine R. Eaton.

**Writing – review & editing:** Katherine R. Eaton, Abigail S. Clark, Nathaniel Hanna Holloway, Kristen Ewen, Erinn M. Muller.

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
