## [Decision Letter · Decision Letter 0]

7 Sep 2022

PONE-D-22-22531A highly effective therapeutic ointment for treating corals with black band diseaseA highly effective therapeutic ointment for treating corals with black band diseasePLOS ONE

Dear Dr. Eaton,

Thank you for submitting your manuscript to PLOS ONE. After careful consideration, we feel that it has merit but does not fully meet PLOS ONE’s publication criteria as it currently stands. Therefore, we invite you to submit a revised version of the manuscript that addresses the points raised during the review process.

We look forward to receiving your revised manuscript.

Kind regards,

Awatif Abid Al-Judaibi, PhD

Academic Editor

PLOS ONE

5. Please amend either the title on the online submission form (via Edit Submission) or the title in the manuscript so that they are identical.

Additional Editor Comments:

Please check the reviewers comments, revise your manuscript and submit the revised version.

Reviewers' comments:

Reviewer's Responses to Questions

**Comments to the Author**

1. Is the manuscript technically sound, and do the data support the conclusions?

Reviewer #1: Yes

Reviewer #2: Yes

2. Has the statistical analysis been performed appropriately and rigorously? 

Reviewer #1: Yes

Reviewer #2: Yes

3. Have the authors made all data underlying the findings in their manuscript fully available?

Reviewer #1: Yes

Reviewer #2: Yes

4. Is the manuscript presented in an intelligible fashion and written in standard English?

Reviewer #1: Yes

Reviewer #2: Yes

5. Review Comments to the Author

Reviewer #1: The authors present a novel pharmacological treatment to inhibit the progression of black band disease on coral in situ in the USVI. The paper is well written and the study is well executed. I have only some minor comments for the consideration of the authors and the editor.

1. Given that this is highly applied research, it could be useful and interesting for the authors to mention some logistical information such as the estimated cost and worker hours required to treat and maintain coral over a given area and period of time. The development of better treatments is important but the development of better application methods is also useful, and only briefly mentioned in the discussion.

2. It could be useful for the authors to briefly mention in the discussion what is known about the species that cause the BBD. Is there a similar complement of bacteria involved in all cases or does it vary greatly and thus justify the broad spectrum treatment?

Reviewer #2: Manuscript: PONE-D-22-22531

Reviewer: Ruba Abdulrahman Ashy

Date: 05/09/2022

The authors have started with a good and clarifying abstract, addressing the summary of their work. They explained their findings well; however, they could strengthen and improve their study with more previous studies on treating the affected corals. As a result, the first paragraph of the discussion was relatively weak. It is better to write a paragraph emphasizing the strengths of their study. However, the study is excellent and interesting.

Below are some queries that need to be clarified.

Q1: Please summarize the main goal and findings agreed with the study hypotheses.

Q2: Please highlight the limitations and the strengths of the study.

Q3: Line (40), it is preferable to write “and” instead of “as well as”

Q4: In line (48), it is preferable to write “focused on” instead of “took a novel approach of focusing on.” Please avoid long run-on sentences.

Q5: In line (51), please avoid the symbols when using them for a text. For example, please write “and” instead of “+.”

Q6: In line (63), please avoid the symbols when using them for a text. For example, please write “and” instead of “+.”

Q7: In line (70) in the introduction, please write “Roseofilum reptotaenium is a common cyanobacterium of BBD ----” instead of “has been described,” as “has been” here means that this species is no longer a pathogenic agent for the BBD.

Also, as it was mentioned above, avoid long phrases.

Q8: In line (76), you began with “While BBD is found globally,” and then, in line (77), you wrote, “In fact, the Caribbean region is described as a coral disease “hotspot.”

Please read the previous studies on the significant occurrence of the BBD in other marine ecosystems, and add these references after the word “globally.” Then, start describing the BBD in the Caribbean regions. This will improve the current study.

Q9: Revise the punctuation in the whole document.

Q10: Lines (122-124), “However, the efficacy of using hydrogen peroxide, carbamide peroxide, or Lugol’s iodine solution has not been tested in the field. Additionally, applying a liquid within a dynamic fluid environment such as shallow coral reefs poses logistical challenges.” Who said that? Write a reference.

Q11: The goals are incomplete. What are your hypotheses?

Please indicate.

Q12: In the methodology, based on which studies were you obtaining your goals, and what are your hypotheses?

Also, did you choose July month based on the previous studies, as you mentioned in the introduction, or on the lagoon of Buck Island Reef National Monument?

Add the temperature when collecting the samples.

Additionally, why did you choose this region and not more?

Q13: Most of the figures need improvement and higher resolution. Figure 1: the map needs to be more precise and have a longitude and latitude. I advise you to use specific software for graphic design to create your figures.

Q14: t is preferable to start the discussion with a more vital beginning, emphasizing that this study is the first to treat the affected corals and agrees with its goals and the previous studies.

Thank you

6. PLOS authors have the option to publish the peer review history of their article (what does this mean?). If published, this will include your full peer review and any attached files.

Reviewer #1: No

Reviewer #2: **Yes: **Ruba Abdulrahman Ashy

---

## [Author Response · Author response to Decision Letter 0]

10 Oct 2022

Within the manuscript, “Table 7” should be called “Table 5”. This was changed in the manuscript in lines 343 and 345.

---

## [Editor Report · Decision Letter 1]

17 Oct 2022

A highly effective therapeutic ointment for treating corals with black band diseaseA highly effective therapeutic ointment for treating corals with black band disease

PONE-D-22-22531R1

Dear Dr. Eaton,

We’re pleased to inform you that your manuscript has been judged scientifically suitable for publication and will be formally accepted for publication once it meets all outstanding technical requirements.

Kind regards,

Awatif Abid Al-Judaibi, PhD

Academic Editor

PLOS ONE

---

## [Editor Report · Acceptance letter]

18 Oct 2022

PONE-D-22-22531R1 

A highly effective therapeutic ointment for treating corals with black band diseaseA highly effective therapeutic ointment for treating corals with black band disease 

Dear Dr. Eaton:

I'm pleased to inform you that your manuscript has been deemed suitable for publication in PLOS ONE. Congratulations! Your manuscript is now with our production department. 

Kind regards, 

on behalf of

Professor Awatif Abid Al-Judaibi 

Academic Editor

PLOS ONE